# Isotopic Traceability (^13^C and ^18^O) of Greek Olive Oil

**DOI:** 10.3390/molecules25245816

**Published:** 2020-12-09

**Authors:** Petros Karalis, Anastasia Elektra Poutouki, Theodora Nikou, Maria Halabalaki, Charalampos Proestos, Effie Tsakalidou, Sofia Gougoura, George Diamantopoulos, Maria Tassi, Elissavet Dotsika

**Affiliations:** 1Stable Isotope Unit, Institute of Nanoscience and Nanotechnology, NCSR Demokritos, 15310 Agia Paraskevi Attiki, Greece; sofi2gr@yahoo.gr (S.G.); g.diamantopoulos@inn.demokritos.gr (G.D.); m.tassi@inn.demokritos.gr (M.T.); e.dotsika@inn.demokritos.gr (E.D.); 2Institute of Geosciences and Earth Resources, Via G. Moruzzi 1, 56124 Pisa, Italy; anastasia_294@hotmail.com; 3Faculty of Pharmacy, Department of Pharmacognosy and Natural Products Chemistry, University of Athens, 15772 Athens, Greece; th-nikou@pharm.uoa.gr (T.N.); mariahal@pharm.uoa.gr (M.H.); 4Department of Chemistry, Food Chemistry Laboratory, National and Kapodistrian University of Athens, Panepistimiopolis Zografou, 15771 Athens, Greece; harpro@chem.uoa.gr; 5Laboratory of Dairy Research, Department of Food Science and Human Nutrition, School of Food and Nutritional Sciences, Agricultural University of Athens, Iera Odos 75, 11855 Athens, Greece; et@aua.gr

**Keywords:** isotopic analysis, ^13^C, ^18^O, greek olive oil origin, traceability, authenticity, biophenols

## Abstract

In recent years, isotopic analysis has been proven a valuable tool for the determination of the origin of various materials. In this article, we studied the ^18^O and ^13^C isotopic values of 210 olive oil samples that were originated from different regions in Greece in order to verify how these values are affected by the climate regime. We observed that the δ^18^O isotopic values range from 19.2 ‰ to 25.2 ‰ and the δ^13^C values range from −32.7 ‰ to −28.3 ‰. These differences between the olive oils’ isotopic values depended on the regional temperature, the meteoric water, and the distance from the sea. Furthermore, we studied the ^13^C isotopic values of biophenolic extracts, and we observed that they have same capability to differentiate the geographic origin. Finally, we compared the isotopic values of Greek olive oils with samples from Italy, and we concluded that there is a great dependence of oxygen isotopes on the climatic characteristics of the different geographical areas.

## 1. Introduction

Olive tree is one of the first fruit trees that were cultivated by humans, both as edible fruit and for oil production. Olive oil is one the main ingredients of the Mediterranean diet, as it prevails over the other edible oils due to its nutritional value and sensorial quality (taste, flavor). Its quality is related not only to the environmental conditions (temperature, sunlight, precipitation) in which the tree grows up but also to cultivation methods (irrigation, fertilization, harvest, storage) [1,2,3,4,5,6] which enhance its commercial value. There is high demand for olive oil, and if this is combined with the high cost of production, it leads to adulteration practices, i.e., by mixing it with lower-quality oils. The adulteration of food products with ingredients of lower quality is a problem that oil suppliers, regulatory agencies, and consumers encounter.

Therefore, the European Union (EU) has proposed two regulations (European Community (EC), No. 2081/91 and 2082/91—Denomination of Protect Origin) in order to guarantee the authenticity of virgin olive oil. These regulations for guaranteeing virgin olive oils that are applied in the EU are known as TSG (Traditional Specialty Guaranteed), PDO (Protected Designation of Origin), and PGI (Protected Geographical Indication), and these connect its quality with its territorial and botanical origin and thus promote its market value.

Hence, many analytical methods are required in order to verify the provenance of high value foods such as olive oil and protect their authenticity. One of the most important methods, in order to determine the authenticity and geographical origin of olive oils, is the Isotope Ratio Mass Spectroscopy (IRMS), as it can assess the authenticity of vegetable products from plants of different photosynthetic pathways [7] and can verify the authenticity and the origin of food sources, e.g., wine [8,9,10].

The carbon isotopic ratio ^13^C/^12^C (expressed as δ^13^C) constitutes a valuable tool for detecting the photosynthetic pathway of the corresponding plants. Plants follow three different biological cycles for the incorporation of CO_2_ during photosynthesis. These are C3 (Calvin cycle), C4 (Hatch–Slack), and CAM (Crassulacean Acid Metabolism), where C3 plants have more depleted δ^13^C values to (−35‰ to −20‰) than the C4 plants (−15‰ to −9‰) [11,12,13,14,15]. The CAM photosynthetic plants have values between the end-members of C3 and C4 plants. Due to the kinetic isotope effect, C4 plants and their metabolites are slightly enriched in ^13^C compared to C3 plant type. As olive tree follows the C3 photosynthetic pathway, it is possible to detect potential additions from different plants. The range of ^13^C isotopic values (−35‰ to −20‰) of C3 plants depend on the ground and rain water, the humidity, and temperature that mainly controls the stomatal aperture [16]. Moreover, ^13^C isotopes combined with ^18^O isotopes analysis can provide information related to the geographical origin of olive oil products. Water pools of the soil resulting by rain water are the main sources of oxygen of plants. Therefore, the δ^18^O values of plants reflect the δ^18^O values of the irrigation water with minor deviations because of atmosphere O_2_ and CO_2_. The ^18^O value of local water is related to climatic conditions, i.e., precipitation, mean temperature, relative humidity of rain, water stress, and geographical characteristics, i.e., latitude, distance from the sea, and altitude, which affect the isotopic composition of precipitation and the transpiration process of leaves and fruit, resulting in the enrichment of the heavy oxygen isotopic contents of plants water compared to groundwater. Thus, the ^18^O/^16^O ratio in plants reflects (a) the isotopic ratio of the ground water sources and precipitation [17], (b) the isotopic fractionation during evapotranspiration [18], and (c) the isotopic exchange between organic molecules and plant water [19]. Hence, if the olive milling is performed in environmental temperatures, and since the milling process causes no enrichment in the δ^18^O of the water, the δ^18^O values of the olives pass in the olive oil, and, consequently, we can differentiate the olive oils produced in different areas.

In this article, we performed oxygen (^18^O) and carbon (^13^C) isotopic analysis to olive oil samples from various regions in Greece. That kind of analysis gives us information about their geographical origin, as the values depend on the climate of each site and help us to differentiate North from South and East from West. Additionally, we measured the δ^13^C isotopic values of the samples’ biophenolic extracts in order to see if we can discriminate the geographical region using these values such as δ^13^C in bulk. Part of these samples was analyzed with LC-MS and FIA-MRMS [20].

These preliminary results encouraged us to extend the measurement to a greater number of olive oil samples coming from different areas to verify the real possibility of their usage for the characterization of geographical origin of olive oils.

## 2. Material and Methods

### 2.1. Sampling

A total of 210 extra virgin olive oil (EVOO) samples were collected from six different olive oil geographical regions of Greece during the harvesting period 2015–2016. All EVOOs are produced from the Koroneiki tree variety, which is the most common and representative olive tree variety in the selected areas. EVOOs were kindly donated by local olive oil producers and cooperatives. Samples were originated from Lasithi, Heraklion (Crete region), Messinia, Lakonia (Peloponnese region), Cephalonia, and Ithaca (Ionian Islands region). In addition, we included olive oil samples that have already been analyzed isotopically from our laboratory; those that originated from Chalkidiki (North Greece region) and Lakonia were collected during the harvesting period 2005–2006, while olive oils from Cephalonia, Messinia, and Lakonia were collected during the harvesting period 2009–2010, raising the total number of samples to 210. Note that Peloponnese and Crete are the main olive oil production areas of Greece, with a significant number of different samples per area, which are harvested and produced with different quality parameters. In this study, the parameters of interest were the production procedure and the used cultivation practice. Ultimately, three different cultivation practices—conventional, integrated, and biological farming—were represented as well as two production procedures: two-phase and three-phase systems. Ionian Islands are not characterized by high olive oil production, although these islands possess different microclimatic conditions from the other selected areas and could reveal interesting results.

All the selected EVOOs were documented with a filling form in order to register all the metadata of each sample such as the virginity index, olive tree variety, production procedure, cultivation practice, microclimatic conditions, and many other parameters that are referred to in the bibliography as factors affecting olive oil quality characteristics. All the acquired data were transferred to databases. After sampling and registration, EVOOs were directly subjected to centrifugation and then transferred to dark, glassy vials; then, a stream of nitrogen was added to remove the air from them and avoid the samples’ oxidation. Finally, the vials were stored at 25 °C.

For biophenols extraction, EVOO samples were elaborated with centrifugal liquid–liquid extraction (CLLE). The extraction procedure was based on the IOC (International Olive Council) proposed protocol [21], which was employed with some modifications for automation purposes due to the significant number of samples intended for extraction. Briefly, 1 g (±0.001) of EVOO was dissolved with 1 mL of n-hexane and homogenized for 3 min in a ratio of 1:1 (*v*/*v*) with an extraction system of methanol and water 8:2 (*v*/*v*). Then, the obtained biphasic system was separated with centrifugation for 3 min at 3000 rpm. The biophenols extracts were defatted twice with n-hexane and evaporated under vacuum conditions and centrifugation at 30 °C, and the dried extracts were stored in glassy vials at −20 °C until analysis day.

### 2.2. Isotopic Analysis

All the (210) olive oil samples were subjected to oxygen (^18^O) versus SMOW (Standard Mean Ocean Water) and carbon (^13^C) versus PDB (Peedee Belemnite) analysis, which were carried out in a Stable Isotope Unit, Institute of Nanoscience and Nanotechnology, NCSR Demokritos (Athens, Greece) on a continuous flow stable isotope mass spectrometer. All measurements followed laboratory standards that were periodically calibrated according to the international standards recommended by the IAEA (International Atomic Energy Agency).

The analysis of stable isotope ^18^O/^16^O ratios of bulk olive oils was performed using isotope ratio mass connected to a pyrolyzer. The analytical conditions are reported in [22]. The ^18^O/^16^O isotopic ratios of water were measured using isotope ratio mass spectrometers connected with a water/CO_2_ equilibration system.

The results are expressed in delta notation (δ) (parts per mille—‰):δ = (R_sample_/R_standard−_1)
where R_sample_ and R_standard_ = ^18^O/^16^O or ^13^C/^12^C ratios of sample and standard, respectively. Measurement precision, based on the repeated analysis of internal standard waters, was 0.5 and 0.2‰ for δ^18^O and δ^13^C, respectively.

## 3. Results and Discussion

The olive oil authentication depends on geographical origin, year of harvest, olive cultivation, and quality. Therefore, in order to evaluate and test the geographical variations of stable isotope fractionation of carbon and oxygen in olive oil, it is necessary to take in consideration precipitation, mean temperature, relative humidity of rain, and water stress.

### 3.1. Regional Climate

Greece has a large variety of climatic conditions but, generally, it can be described as Mediterranean, with dry and hot summers and wet mild winters. The presence of the Pindos ridge and other mountains that cross the mainland partition the country in two parts, west and east, and they are responsible for the existence of continental climate areas with year-round precipitation and rather cold winters. These mountains induce the orographic uplift phenomenon according to which the maximum precipitation is concentrated on the western part of the country. Specifically, during the winter, air masses that originated from the Atlantic Ocean or western and central Mediterranean are forced over the mainland, collide with the mountains, and are uplifted and cooled, producing large amounts of rainfall across western Greece and smaller amounts over central and eastern Greece. These air masses, on their way east, cross the warm Aegean Sea and produce some heavy rains in the Eastern Aegean islands and the coast of Turkey. During summer, the driest period of the country, there are some frontal rainfalls in Northern Greece and local thermal storms that contribute to significant amounts of precipitation in mountainous areas. In southern and central Greece, 70–80% of the annual precipitation is measured between October and April. Furthermore, in northern Greece, there is a second maximum of precipitation with uniform distribution. A temperature gradient, from subtropical (south Greece) through temperate to cold (north Greece), is observed. The annual average temperature ranges from 19.7 °C in Ierapetra (35.0 N), 17.7 °C in Athens (38.0 N), and 15.7 °C in Thessaloniki (40.5 N) to 11.7 °C in Karpenisi (38.9 N). In Table 1, one can see the annual climatic parameters (temperature, precipitation, humidity) for the years and regions of olives harvesting.

These climatic conditions are reflected in the isotopic ratio of rain water. The most negative precipitation isotopic values are from northern Greece, i.e., Macedonia, Eastern Macedonia, and Thrace, while the more positive values come from south Greece, Sterea, Peloponnese, and Crete Island [23,24]. This is attributed to the air temperature gradient and the massifs of the country. One of the factors that can be identified as influential is the distance of the stations from the sea. The mean δ^18^O isotopic values of precipitation are lower in continental stations than the ones in coastal and island. Isotopic values of precipitation at higher altitudes are more negative than the ones at lower altitudes. Additionally, due to the presence of the Pindos ridge between West and East Greece, the clouds are discharged over the west part of Greece and as result, only isotopically depleted precipitation reaches the eastern part.

### 3.2. Isotopic Values

In Table 2 we present the δ^13^C and δ^18^O mean values of olive oils produced in different areas of Greece, which can be differentiated from one another by their geographical location. The four locations are (a) North Greece, including Chalkidiki (CHA), (b) Peloponnese, including Messinia (MES) and Lakonia (LAK), (c) Ionian Islands, including Cephalonia (KEF), Ithaca (ITH) and (d) Crete, including Lasithi (LAS) and Heraklion (HER).

Generally, for all samples, data ranges from −32.7‰ to −28.3‰ for δ^13^C and from 19.2‰ to 25.2‰ for δ^18^O. Specifically, Chalkidiki has δ^13^C values from −29.8‰ to −29.1‰ with a mean value of all the years of harvest of −24.4‰ and δ^18^O values from 23.1‰ to 23.9‰, mean value 23.5‰; Messinia from −31.6‰ to −28.5‰ and δ^18^O values from 21.4‰ to 22.7‰ with mean values of all the years of −29.4‰ and 22.6‰ respectively; Lakonia presented δ^13^C values from −31.6‰ to −28.3‰ and δ^18^O values from 23.2‰ to 23.8‰ with mean values of all the years of −29.2‰ and 24.3‰; Cephalonia presented δ^13^C values from −32‰ to −28.3‰ and δ^18^O values from 20.3‰ to 21.7‰ with mean values of all the years −29.4‰ and 21.3‰, respectively; Ithaca presented δ^13^C values from −32.2‰ to −30.3‰ and δ^18^O values from 20.1‰ to 22‰, mean values −31.1‰ and 21.4‰ respectively; Lasithi presented δ^13^C values from −31.5 ‰ to −28.2‰ and δ^18^O values from 23.6‰ to 24.9‰ with mean values −29.7‰ and 24.3‰ respectively; Heraklion presented δ^13^C values from −31.4 ‰ to −28.2‰ and δ^18^O values from 23.7‰ to 24.9‰ with mean values −29.9‰ and 24.1‰, respectively.

All the isotopic values present an increment from North (CHA; 2005–2006) to South Greece (LAK; 2005–2006) and from West (Ionian Islands; 2015–2016) to East (Crete; 2015–2016) probably due to isotopic composition of ground and rain water, which is relative to humidity and temperature. Most of the samples collected in different years show δ^13^C values ranging between −31.6‰ and −28.3‰ for Lakonia (Figure 1), indicating a discrimination of the Calvin biosynthetic process in the olive fruits and the influence of environmental and physiological factors (humidity, temperature, availability of water, water stress, ripening stage of the olive, and plant age), which control stomatal aperture and the internal CO_2_ concentration in the leaf. Furthermore, the ^13^C values of oil samples coming from adjacent areas (MES, LAK) (2015–2026) and from different years were found to be very similar with the harvesting year (ex. MES, 2009 and 2016) appearing to have no remarkable effects on carbon isotope discrimination. As shown in Figure 1, four locations (2015–2016) (Crete, MES, LAK, CEPH) have similar δ^13^C values (−30 ± 1‰), contrary to the δ^18^O values (Figure 2) that show a greater variability, ranging from 20.3‰ to 24.7‰. The δ^18^O values of samples originating from Crete (Figure 2) (2015–2016) present an enrichment in δ^18^O values compared to those produced in the other areas (CHA, MES, LAK, CEPH, and ITH) of Greece. Messinia and Lakonia samples, collected in different years, show oxygen values (Figure 2; Table 2) higher than the ones from Ionian Islands, and it appears that oxygen isotopes composition is related to climate of the olive oil production area, evidencing their relation to the geographic locations. The Ionian Islands are in a unique geographical location, which isolates them from the mainland because of Pindos and Taygetos Mountains, which stop the cold climate coming from the west. Therefore, the Ionian Islands are characterized by humidity and high precipitation, making their climate different from other areas in Greece. Messinia and Lakonia are characterized by higher temperatures and less precipitation than Ionian Islands, which are reflected in the δ^18^O values.

Thus, we observe that the δ^18^O values differ among olive oils produced in the north (Chalkidiki), west (Ionian Islands, Messinia), and southern regions (Crete) located near the sea. This isotopic enrichment from North (CHA) to South (Crete) and from West (MES, Ionian Islands) to East (Crete) Greece is related to dryness of the climate and to temperature.

In Figure 3 and Figure 4, the δ^13^C and δ^18^O values of olive oils from Cephalonia, Messinia, and Lakonia are compared between olive oil production years 2009–2010 and 2015–2016. By comparing the data of the two years, a great differentiation in both isotopes is observed. The different climatic conditions between the two years can justify these differences. According to the Hellenic National Meteorological Service, the mean temperature of Cephalonia for the year 2009–2010 was 19.2 °C and the mean annual precipitation was at 710 mm, but for the year 2015–2016, the mean temperature was 18.7 °C, and the mean annual precipitation was at 900 mm. For Messinia, in 2009–2010, the mean temperature was 18.8 °C and the mean annual precipitation was 584 mm, and for 2015–2016, 18.0 °C was the mean temperature and 800 mm was the mean annual precipitation. Finally, for Lakonia in 2009–2010, the mean temperature was 19.1 °C, and the mean annual precipitation was 450 mm, while for year 2015–2016, these values were 18.4 °C and 700 mm, respectively. In all cases, a higher temperature and drier climate increases both isotopic values.

It is worth noting that whatever the year of production of the olive oil, the oils from Sparta (LAK) show oxygen enrichment compared to those of Messinia (MES). The δ^18^O values differ among olive oils that were originated from fruits grown in the west part (Messinia) of Greece, near the sea and from the plain of Sparta (Lakonia), which is surrounded by high mountains with only one exit to the sea, where higher temperatures are recorded during the summer. Sparta has significantly higher average summer temperatures than any Greek region for each period. The differences are so distinguished for Sparta (Lakonia) to the point that there are months with more than 1 degree Celsius higher temperature from all the other parts of Greece. According to the data of EMY (the Hellenic National Meteorological Service) for the period 1974–2004 (30 years), the average maximum temperatures in June, July, and in August, for Sparta, are 32.2, 34.8, and 34.3 °C, respectively. The average maximum temperature for the 8 years from 2009 to 2016 in the warmest month is 36 °C, which is the highest in the territory. Characteristic is the average maximum temperature of July 2012, when Sparta set the national record for the highest average with 38.3 °C. In addition, Sparta has average maximum temperatures of 0.5 degrees Celsius higher than Messinia during the entire summer and has also the highest frequency (an average of more than 4 days per year) in temperatures above 40 degrees Celsius from all public stations in Greece.

Indeed, the geographical location of Sparta (Lakonia), the geomorphology of the Evrotas valley, the distance of the city from the sea, and its latitude (located a little further south than Messinia) are factors that support the very hot summers that the city experiences. On the one hand, the fact that the plain of Sparta is surrounded by the mountains of Taugetos and Parnonas acts protectively against the sea breeze and makes it an ideal place for the development of warm downhill winds from almost any direction. Therefore, this topography and high temperature are reflected in the isotopic values. In fact, spring water from the Lakonia plain is more enriched in ^18^O and ^2^H in relation to the other areas in Greece, and this is reflected in a decrease in slope and deuterium excess showing that this is affected by the influence of evaporation processes [23,24].

Finally, Sparta and Crete present the highest isotopic values of spring water compared to the mainland of Greece, indicating a climatic transition toward warmer and drier conditions. Additionally, the highest values of δ^18^O of spring water of Crete, which is located near the sea, indicates a possible contribution of vapor derived by evaporation of the sea surface, and this is also reflected in the ^18^O isotopic values of olive oil.

#### Relation between δ^18^O Water and δ^18^O Olive Oil

The isotopic analysis of ^18^O of water and olive oil originated from four different areas of Greece in Peloponnese (MES and LAK) and South Greece (Crete-HER and LAS) are given in Table 3. These water and olive oil samples are special “markers” for which we know the irrigation water of the plants. The relationship between the ^18^O isotopic composition of olive oils and that of the local water is evident and reflects the effect of climate and of the evapotranspiration processes in olive plants.

Considering that the major factor that affects the oxygen isotopic composition of olive oil is the irrigated water, the observed variation of the isotopic composition of oil is modulated by the climatic conditions of each plant: the apparent differences of the oxygen isotopic composition of samples can be translated mainly into environmental conditions, as latitude, altitude, temperature, and distance from the sea.

The equation between δ^18^O_oil_ and δ^18^O_w_, that incorporates only the effect of irrigated water on the isotopic composition of olive oil, is:

δ^18^O_oil_ = 0.8081δ^18^O_w_ + 28.103 with r^2^ = 0.98 and number of samples *n* = 150.

From the measured δ^18^O_w_ values and with the use of the equation, we could calculate the δ^18^O_oil_ of olive oil values. Similarly, we calculated the δ^18^O of the olive oil with the δ^18^O_w_ datasets of the literature [23,24].

### 3.3. Biophenolic Extracts of Olive Oils

In Table 4, the δ^13^C isotopic values of biophenolic extracts of olive oils originated from Messinia, Lakonia, Ithaca, and Heraklion are compared with the isotopic values of bulk olive oils from the same region. The δ^13^C values in bulk oil were significantly correlated with those of biophenolic extracts. Biophenolic extract of Heraklion has a δ^13^C isotopic value of −30.1‰ with the corresponding value of the bulk olive oil to be −29.9‰; for Ithaca, the respective values are −31.0‰ and −31.1‰; for Lakonia, the corresponding values are −29.3‰ and −29.9‰; and for Messinia, the respective values are −29.2‰ and −30.2‰. For Ithaka and Heraklion, the isotopic values of biophenolic extracts and bulk olive oil do not show any statistically important difference, but for Lakonia, there is a positive effect for the biophenolic extract of 0.6‰ and for Messinia, there is a positive effect of 1.0‰. The mean difference confirms previous results [22,26] and shows that the δ^13^C of biophenolic extracts have same capability to differentiate the geographic origin as δ^13^C of bulk. These observations are in agreement with the findings of a recent study where other analytical techniques were employed and specifically LC-HRMS and FIA-MRMS [20].

### 3.4. Isotopic Values from Olive Oil Samples Produced in Other Mediterranean Areas

In order to clarify the effect of the different geographical origin in isotope values of olive oil products and to differentiate Greek olive oil from other imported olive oil products, the δ^13^C ‰ V-PDB versus δ^18^O ‰ V-SMOW diagram (Figure 5) was constructed. In the same figure, Italian olive oil samples [22,25] with different geographical origins, from Trentino, Tuscany, and Sicily are also plotted. In this diagram, it is apparent that the Italian olive oils follow the trend of North–South enrichment of oxygen isotope values more clearly than Greek olive oils, which is probably due to many years of documentation of olive oils and also due to the geographical span of Italy (greater than Greece). Italy, geographically, has a length from north to south of 1300 km, in contrast to Greece that has a length of 650 km from north to south. Thus, the climate of Italy varies from north to south, since the north side is dependent on the rainwater and low temperatures of the Alps, and on the south side, Sicily, an island in the center of the Mediterranean, has high temperatures and dry weather. Although Greece has different climates that vary throughout its geographic region, this variation cannot be compared with that of Italy. The δ^18^O enrichment is not surprising, because it is consistent with the oxygen relation to the geoclimatic parameter. Greece, with a temperate climate, shows ^18^O isotopic values intermediate of those of Trentino and Sicily: lower than Sicily and higher of Trentino, which is characterized by a low temperature and wet climate.

## 4. Conclusions

In this article, we performed carbon and oxygen isotope analysis in 210 olive oils samples with different geographical origin in order to identify if there are differences among them due to their geographical diversity. We observed that olive oils from Ionian Islands can be differentiated from olive oils originated from Crete and Chalkidiki according their δ^18^O and their δ^13^C isotopic values. Specifically, for oxygen, the isotopic values range between 19.2 ‰ and 25.2 ‰, and for carbon, they range from −32.7 ‰ to −28.3 ‰. Thus, it is concluded that isotopic values are strongly affected by the meteoric water and the climate regime of each area, such as the temperature and the distance from the sea. Specifically, the vicinity to the sea and the dryness of the climate are mainly positively related factors to the isotope values in contrast to latitude, which is negatively related. Additionally evident is the fact that samples from the same region can differ isotopically between different cultivation periods due to the different weather conditions of each year. Finally, we compared our results with the results from Italy and observed that although they are in relative close geographical proximity, each has its own isotopic composition affected by climate factors and geographical parameters and, thus, they can be used as a discrimination tool for olive oil products.

## Figures and Tables

**Figure 1 molecules-25-05816-f001:**
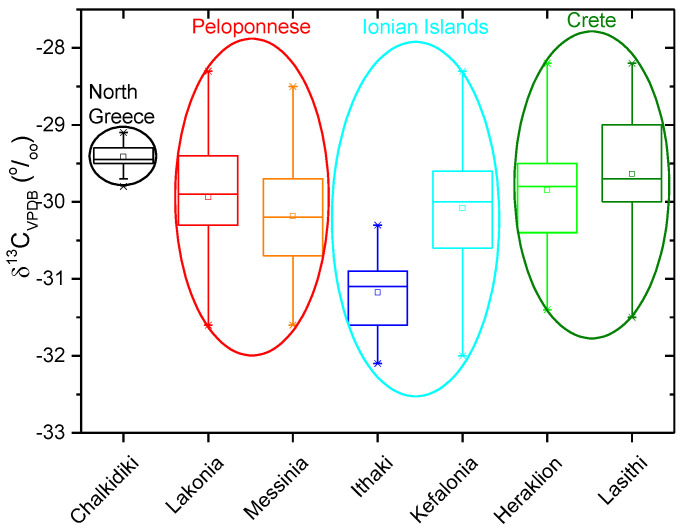
δ^13^C_VPDB_ of olive oils from Greek areas.

**Figure 2 molecules-25-05816-f002:**
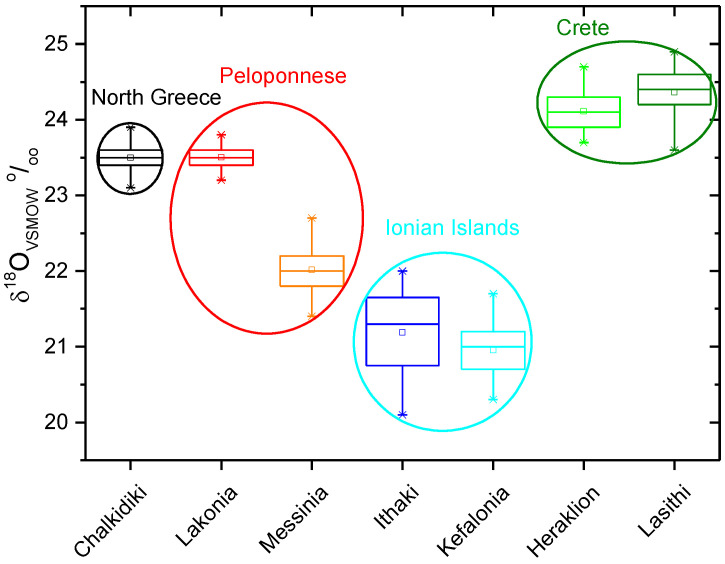
δ^18^O_VSMOW_ of olive oils from Greek areas.

**Figure 3 molecules-25-05816-f003:**
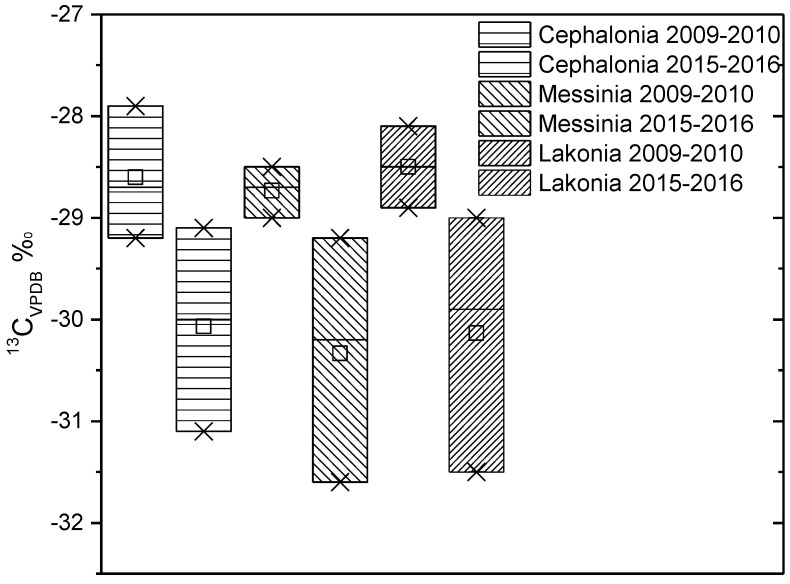
Comparison of δ^13^C_VPDB_ values of olive oils from Cephalonia, Messinia, and Lakonia between years 2009–2010 and 2015–2016.

**Figure 4 molecules-25-05816-f004:**
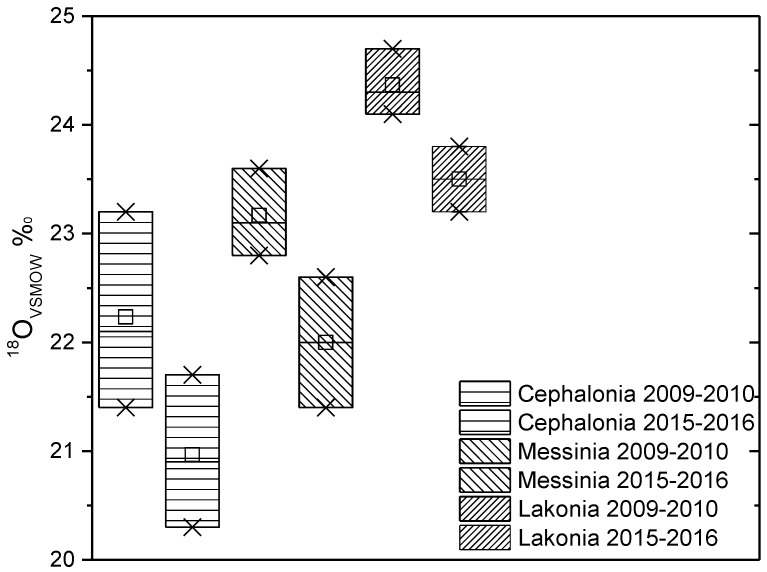
Comparison of δ^18^O_VSMOW_ values of olive oils from Cephalonia, Messinia, and Lakonia between years 2009–2010 and 2015–2016.

**Figure 5 molecules-25-05816-f005:**
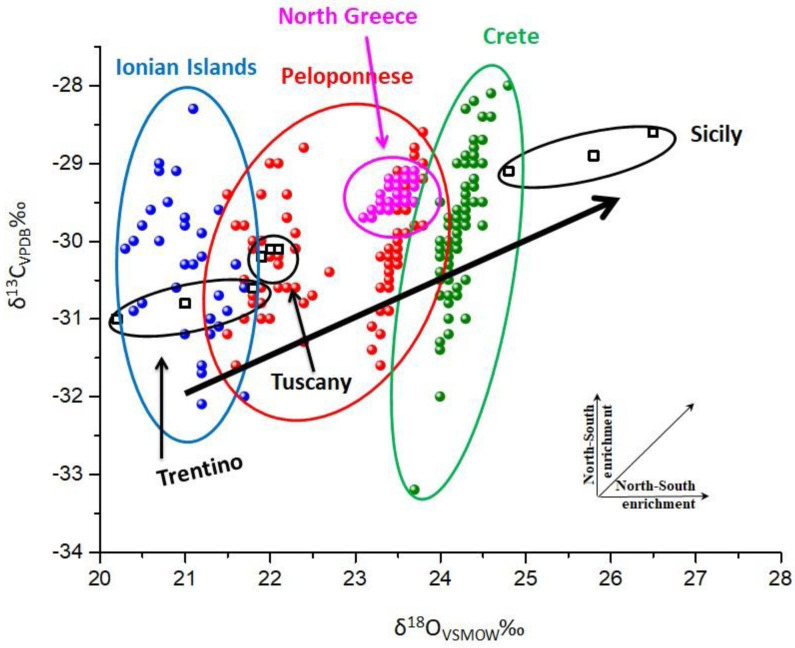
δ^13^C_VPDB_ ‰ versus δ^18^O_VSMOW_ ‰ for bulk olive oil samples.

**Table 1 molecules-25-05816-t001:** The climatic parameters of the olive oil cultivation area in Greece.

**Precipitation (mm)/Year**							
**Year of Harvest**	**Chalkidiki** **(CHA)**	**Messinia** **(MES)**	**Lakonia** **(LAK)**	**Ithaca** **(ITH)**	**Cephalonia** **(CEPH)**	**Lasithi** **(LAS)**	**Heraklion** **(HER)**
2005 *	140		629				
2006 *	206		269				
2010		594	500		760		
2015–2016		700	700	>1000	800–1000	400–600	400–600
**Temperature (°C)**							
**Year of harvest**	**Chalkidiki**	**Messinia**	**Lakonia**	**Ithaca**	**Cephalonia**	**Lasithi**	**Heraklion**
2005 *	16.2		17.9				
2006 *	16.3		18				
2010		18.5	18.5		19		
2015–2016		18	18	17.8	18	19	19
**Relative Humidity (%)**							
**Year of harvest**	**Chalkidiki**	**Messinia**	**Lakonia**	**Ithaca**	**Cephalonia**	**Lasithi**	**Heraklion**
2005 *	60		59				
2006 *	60		60				
2010							
2015–2016							

* From reference [25].

**Table 2 molecules-25-05816-t002:** The δ^13^C_VPDB_ and δ^18^O_VSMOW_ isotopic values of Greek olive oils expressed as mean values.

Year of Harvest (YH)	Method		N. Greece (N^o^of Samples)	Peloponnese (N^o^ of Samples)		Ionian Islands (N^o^ of Samples)		Crete (N^o^ of Samples)	
			CHA ^b^	MES ^c^	LAK ^d^	CEPH ^e^	ITH ^f^	LAS ^g^	HER ^h^
YH ^a^, *2005*	δ^13^C‰ vs. V-PDB	mean	−29.3 (20)		−28.4 (11)				
		SD	0.6		0.6				
	δ^18^O‰ vs. V-SMOW	mean	23.4 (20)		25.1 (11)				
		SD	0.6		0.7				
YH ^a^, *2006*	δ^13^C‰ vs. V-PDB	mean	−29.4 (20)		−28.3 (17)				
		SD	0.2		0.4				
	δ^18^O‰ vs. V-SMOW	mean	23.5 (20)		24.1 (17)				
		SD	1.5		0.3				
YH, *2009*	δ^13^C‰ vs. V-PDB	mean		−28.7 (30)	−28.5 (25)	−28.7 (25)			
		SD		0.3	0.5	0.4			
	δ^18^O‰ vs. V-SMOW	mean		23.1 (30)	24.3 (25)	22.1 (25)			
		SD		0.5	0.4	0.5			
YH, *2015–2016*	δ^13^C‰ vs. V-PDB	mean		−30.2 (38)	−29.9 (53)	−30.0 (24)	−31.1 (10)	−29.7 (38)	−29.9 (47)
		SD		0.4	0.6	0.6	0.9	0.9	0.9
	δ^18^O‰ vs. V-SMOW	mean		22.0 (38)	23.5 (53)	20.9 (24)	21.4 (10)	24.3 (38)	24.1 (47)
		SD		0.6	0.2	0.3	0.8	0.7	0.5

^a^ From reference [25]; ^b^ CHA: Chalkidiki; ^c^ MES: Messinia; ^d^ LAK: Lakonia; ^e^ CEPH: Cephalonia; ^f^ ITH: Ithaca; ^g^ LAS: Lasithi; ^h^ HER: Heraklion.

**Table 3 molecules-25-05816-t003:** Oxygen isotopic composition of olive oil and of spring water samples from the same regions. Peloponnese (MES and LAK) and South Greece (Crete-HER and LAS).

Location	δ^18^O_VSMOW_ of Water	δ^18^O_VSMOW_ of Olive Oil	N^o^ Samples
MES	−7.0	22.5	35
LAK	−5.5	23.5	43
HER	−5.0	24.1	37
LAS	−4.8	24.3	35

**Table 4 molecules-25-05816-t004:** The mean δ^13^C_VPDB_ isotopic values of whole Greek olive oils and their biophenolic extracts.

Year of Harvest	Method	Olive Oil		Peloponnese (N^o^ of Samples)		Ionian Islands (N^o^ of Samples)	Crete (N^o^ of Samples)
				MES	LAK	ITH	HER
2015–2016	δ^13^C‰ vs. V-PDB	Bulk olive oil	mean	−30.2 (38)	−29.9 (53)	−31.1 (10)	−29.9 (47)
			SD	0.4	0.6	0.9	0.9
	δ^13^C‰ vs. V-PDB	Biophenols extract	mean	−29.2 (38)	−29.3 (53)	−31.0 (10)	−30.1 (47)
			SD	0.4	0.5	0.8	0.7
	Mean difference			−1.0	−0.6	−0.1	+0.2

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
