# Peer review of "Isotopic Traceability (13C and 18O) of Greek Olive Oil"

_molecules, 2020, doi:10.3390/molecules25245816_

Round 1

Reviewer 1 Report

  1. Line 2, in title it is not appropriate to use symbols (13C +18O), I suggest to use “and” or comma “,”.
  2. In “Abstract” section, please add some specific data (e.g. variation range for δ13C and δ18O), it is a little bit too generally. Line 17, change “many olive oil samples” by “210 olive oil samples”.
  3. Line 107. Please add an additional sentence for a better understanding (e.g. “a stream of nitrogen was added to remove the air from vials and avoid samples oxidation and then the vials were stored at 25°C).
  4. Equation in the line 130 has been deprecated since 2011 (Coplen RCM 25, 2538-2560). Please, modify it.
  5. There are more than a space between two word (please check through text) (e.g. line 42, 69, 129, 299, 311).
  6. In figures 1, 2, 3, 4, 5 and 6, please add in the axis titles, as subscript, the international standards versus the measurements were done (e.g. δ18OVSMOW and δ13CVPDB). The same observation for table 3 (where we must add also “δ” in front of 18O).
  7. Line 332, please change “two hundred and ten” by “210”.
  8. Lines 334 and 335, please delete “‰”.

Author Response

Dear reviewer,

     We would like to thank you for your comments that helped us to improve our manuscript with title “Isotopic Traceability (13C and 18O) of Greek Olive Oil”. We submit the revised version in which we made the appropriate corrections. All the changes are explained point by point below.

Reviewer 1

Comment 1: Line 2, in title it is not appropriate to use symbols (13+18O), I suggest to use “and” or comma “,”.

Answer 1: We changed Line 2 as suggested i.e. instead of symbol “+” we used “and”.

Comment 2: In “Abstract” section, please add some specific data (e.g. variation range for δ13C and δ18O), it is a little bit too generally. Line 17, change “many olive oil samples” by “210 olive oil samples”.

Answer 2: In Line 17 we changed “many” by “210”.

In Lines 19-11 we changed the sentence “We observed that the differences between the olive oils’ isotopic values depended…” as follows “We observed that the δ18O isotopic values ranges from 19.2 ‰ to 25.2 ‰ and the δ13C from -32.7 ‰to -28.3 ‰. These differences between the olive oils’ isotopic values depended…”

Comment 3: Line 107. Please add an additional sentence for a better understanding (e.g. “a stream of nitrogen was added to remove the air from vials and avoid samples oxidation and then the vials were stored at 25°C).

Answer 3: In line 107 we changed the sentence “…glassy vials and stored under nitrogen conditions at 25oC, for stability purposes.” As follows “…glassy vials and then a stream of nitrogen was added to remove the air from them and avoid samples oxidation. Finally, the vials were stored at 25°C.”

Comment 4: Equation in the line 130 has been deprecated since 2011 (Coplen RCM 25, 2538-2560). Please, modify it.

Answer 4: In line 120 we changed the sentence “All the (210) olive oil samples were subjected to oxygen (18O) and carbon (13C)”. As follows “All the (210) olive oil samples were subjected to oxygen (18O) versus SMOW and carbon (13C) versus PDB”. In line 129 we change the sentence “The results are expressed in standard delta notation (δ) as per mil (‰) deviation from the standards V-SMOW for δ18O   and PDB for δ13C as:”, as follows “The results are expressed in delta notation (δ) (parts per mille – ‰):”

 In line 130 we changed the equation “δ=((Rsample-Rstandard)/Rstandard)*1000” in order to be coherent with note 9 and 10 p. 2555 of Coplen RCM 25, 2538-2560, as follows “δ=(Rsample/Rstandard - 1)”

Comment 5: There are more than a space between two word (please check through text) (e.g. line 42, 69, 129, 299, 311).

Answer 5: We checked the text and corrected all the double spaces.

Comment 6: In figures 1, 2, 3, 4, 5 and 6, please add in the axis titles, as subscript, the international standards versus the measurements were done (e.g. δ18OVSMOW and δ13CVPDB). The same observation for table 3 (where we must add also “δ” in front of 18O).

Answer 6: We added in the axis titles and captions of figures 1-6 the international standards versus our measurements were performed “VSMOW” and “VPDB”.

In Table 3 we added “δ” in front of 18O and subscript “VSMOW”.

In the caption of Tables 2 and 4 we added the subscripts “VSMOW” and “VPDB” in δ18O and δ13C respectively.

Comment 7: Line 332, please change “two hundred and ten” by “210”.

Answer 7: In line 335 we changed “two hundred and ten” by “210”

Comment 8: Lines 334 and 335, please delete “‰”.

Answer 8: In lines 334 and 335 we deleted the “‰”.

Yours sincerely,

 P. Karalis

Reviewer 2 Report

The manuscript aims at characterizing olive oils produced in different geographical areas (mainly Greece). 

The study is well planned from the analytical point of view unfortunately, authors draw conclusions about discrimination about samples, even if data analysis is completely missing. 

I personally think the authors have two options: 

1) reorganize the manuscript slightly changing its aim. If the aim is characterizing oil from different geographical areas on the basis of isotopic analysis, they should remove the last part where they discuss grouping tendencies among samples

2) Apply a classifier, or, at least, PCA on data and discuss grouping tendencies among samples. 

At its current form, the manuscript is not acceptable for publication

Author Response

Dear reviewer,

     We would like to thank you for your comments that helped us to improve our manuscript with title “Isotopic Traceability (13C and 18O) of Greek Olive Oil”. We submit the revised version in which we made the appropriate corrections. All the changes are explained point by point below.

Reviewer 2

Comments:The manuscript aims at characterizing olive oils produced in different geographical areas (mainly Greece). The study is well planned from the analytical point of view unfortunately, authors draw conclusions about discrimination about samples, even if data analysis is completely missing. I personally think the authors have two options: 

1) reorganize the manuscript slightly changing its aim. If the aim is characterizing oil from different geographical areas on the basis of isotopic analysis, they should remove the last part where they discuss grouping tendencies among samples

2) Apply a classifier, or, at least, PCA on data and discuss grouping tendencies among samples. 

Answer: We followed option 1 thus we removed Fig 6 and the paragraph “In Figure 6 Greek olive oils are isotopically compared with olive oils originated from other Mediterranean countries (Italy, Morocco, Tunisia, Turkey, Portugal and Spain) [22,25,27-29].Three groups can be observed in this diagram according to the isotopic values. Samples from Morocco and Portugal constitute the first group which is the one with the most positive δ18Ο values. The second group consists of Spain, Tunisia and Turkey while the third one consists of olive oils originated from Greece and Italy. These values, again, indicate the great dependence of oxygen and carbon isotopes on the climatic characteristics of the different geographical areas.” that was referred to the grouping tendencies among olive oils of Mediterranean area.

In abstract we changed:

In Line 24 we changed the sentence “Finally, we compared the isotopic values of Greek olive oils with samples from Mediterranean countries”, as follows “Finally, we compared the isotopic values of Greek olive oils with samples from Italy”.

In conclusions we changed:

In Line 326 we added the sentence “origin in order to identify if there are differences among them due to their geographical diversity.”

In Line 329 we added the sentence “Specifically for oxygen the isotopic values range between 19.2 ‰ and 25.2 ‰ and for carbon from -32.7 ‰to -28.3 ‰.”

In Line 336 we changed “Mediterranean” as follows “Italy”

In references we removed 27, 28, 29 because they refer to the deleted part.

Yours sincerely,

P. Karalis

Round 2

Reviewer 2 Report

The author modified the manuscript according to the reviewers' comments, and I personally think it is now ready for acceptance